# Unexpectedly complex distribution pattern of chestnut pest *Niphades castanea* Chao (Coleoptera: Curculionidae) based on mtDNA and ITS markers

Bin Mao [ORCID]◉, Mi Shen◉, Yue Fu, JiaXin Wang, Peng Yu, YunLi Xiao [ORCID]*

Hubei Key Laboratory of Economic Forest Germplasm Improvement and Resources Comprehensive Utilization, Hubei Collaborative Innovation Center for the Characteristic Resources Exploitation of Dabie Mountains, College of Biology and Agricultural Resources, Huanggang Normal University, Huanggang City, Hubei, China

◉ These authors contributed equally to this work.

* xiaoyunli0817@126.com

**Data Availability Statement:** All these nucleotide sequences were deposited in GenBank (https://www.ncbi.nlm.nih.gov/genbank/) under the accession numbers for 167 COI (accession

## Abstract

*Niphades castanea* Chao (Coleoptera: Curculionidae), an important fruit insect pest of chestnuts (*Castanea* spp.), could cause chestnut involucre abscission ahead of time through larvae boring and feeding basal involucres, eventually causing huge economic losses. In this research, mitochondrial (COI and COII) and nuclear (ITS1) markers were used to investigate genetic variation among 15 different geographical populations of chestnut pest *N castanea*. The molecular diversity of *N. castanea* populations revealing three main phylogenetic clusters, with cluster I specifically distributed at high elevations in the western sampling points. Mitochondrial genes indicated population expansion events, and the ITS1 marker suggested a history of population expansion. Genetic diversity differentiation was significant among populations, indicating that geographical isolation impacts genetic differentiation among these places. AMOVA analyses confirmed substantial genetic differentiation between populations. Mantel correlogram analyses revealed a significant positive correlation between genetic differentiation and altitude/geographical distance at lower elevations and ranges, which reversed to a negative correlation at higher altitudes and ranges for all markers, indicating the role of altitude and geographical distance in shaping genetic diversity in *N. castanea*. This study contributes to a comprehensive understanding of the distribution, genetic diversity, and evolutionary history of *N. castanea* in the central of China, underscoring the impact of geographical factors on its genetic structure.

## Introduction

Chestnuts, belonging to the Fagaceae family and represented by species such as *Castanea spp.*, are indigenous to the deciduous forests of Asia, eastern North America, and Europe [1, 2]. These trees hold significant ecological and economic importance, contributing substantially to

number: PP526771-PP526937), 109 COII (accession number: PP546395-PP546503), and 137 ITS1 (accession number: PP542683-PP542819) sequences. Additional accession numbers can be found in the Supporting Information files of this manuscript (S4 Table).

**Funding:** This study was supported by the National Natural Science Foundation of China (NSFC), grant numbers 31702048 (to Y.L.X.)."

**Competing interests:** The authors have declared that no competing interests exist.

ecosystems and yielding nutritious nuts that sustain wildlife and provide specialty foods for humans [3]. The nuts from *C. mollissima* and *C. henry*i are favored in East and Southeast Asia due to their high content of starch, soluble sugars, proteins, amino acids, minerals, and fiber, along with vitamins and low-fat levels [4, 5]. In China's rural areas, chestnut production is a vital source of income, with many people relying on nut sales as their primary economic resource. *Niphades castanea* Chao, a significant pest of *C. mollissima*, *C. segllinii* Dode, and *C. henryi*, can prematurely cause the abscission of chestnut involucres by larvae boring and feeding on the basal involucres [6].

Molecular phylogeographic studies could provide valuable insights to explore genetic structure, prevalent time scales, geographic modes, and even demographic history [7, 8]. Species-specific life history characteristics can influence geographic population structure. *N. castanea* possesses a remarkable ability to adapt to environmental stressors, leading to substantial economic losses in China. Additionally, the potential for chestnut pests to hide within the fruits during trade and transportation could spread them to other regions, thereby posing a threat to the ecological environment and agricultural production in these new areas. Moreover, the spatial genetic structure of populations may reflect the adaptive geographic variation and adaptive potential to respond to environmental changes in species. Up to now, the phylogeographic patterns of a list invertebrate and vertebrate species have been well studied [9–11]. The mitochondrial protein-coding gene, namely cytochrome c oxidase subunit I or II (COI or COII) and nuclear Internal transcribed spacer1 (ITS1) were import for DNA barcoding of all living species [12–14]. Although the complete mitochondrial genome of the chestnut pest *N. castanea* has been sequenced [15], the phylogeographic patterns within this species remain poorly understood. Furthermore, there is a lack of intra-specific phylogeographic information available for *N. castanea*.

In this study, the geographical pattern of genetic variation for *N. castanea* was assessed based on both ITS1 and mitochondrial markers (COI and COII). Moreover, geographical pattern of haplotypes of *N. castanea* were also examined. Genetic diversity was analyzed within and among different *N. castanea* populations sampled in China, hoping to supply evidence for developing scientific strategies for prevention and control of *N. castanea* in China's central regions.

## Materials and methods

### Materials sampling

*N. castanea* populations were investigated all over China between 2019 and 2021 The linear geographical distance among the sampling locations spanned from 22.984˚N to 40.266˚N latitude and from 97.892˚E to 121.054˚E longitude, with altitudes varying from 35.9 meters to 1595 meters. The sampling sites included a broad range of regions across China, such as Anhui Province, Beijing City, Gansu Province, Guangxi Province, Guizhou Province, Hebei Province, Henan Province, Hubei Province, Hunan Province, Jiangsu Province, Jiangxi Province, Shandong Province, Shanxi Province, Shaanxi Province, Sichuan Province, Tianjin City, Yunnan Province, and Zhejiang Province, as detailed in our previous review [16]. Totally, 179 *N. castanea* individuals were frozen alive and stored at -80˚C until further DNA extraction (S1 Table). Specimens have been deposited at Huanggang Normal University (Hubei province, China).

### Genomic DNA extract, amplification, and sequencing

Total genomic DNA was extracted and purified from single viscera-removed larva using DNeasy Blood and Tissue Kit (Qiagen, Germany). Abdomen of each sample was removed

prior to DNA extraction. Quality of *N. castanea* DNA isolates were verified in 1% agarose gel stained by GelRed (Biotium, USA), as well as quantified by spectophotometer (NanoDrop 1000, Thermofisher Scientific, USA). Fragments of mtDNA genes (COI and COII) and nuclear ITS1 region were served as molecular markers in this research.

Primers LCO1490/HCO2198 were used to amplify COI region [17]. The following primers were newly designed: 5'-AAT ATG GCA GAA TAG TGC AA-3' (forward) and 5'-TGG TTT AAG AGA CCA TTA CTT G-3' (reverse) for COII; 5'-ACA CAC CGC CCG TCG CTA CTA-3' (forward) and 5'-ATG TGC GTT CRA AAT GTC GAT GTT CA-3' (reverse) for ITS1. PCR amplifications were performed in 15 µL volumes, containing 1.5 µL of 10×PCR buffer, 0.2mM each dNTP, 0.2 µM each of primer, 0.25 U Taq DNA polymerase (Takara, Dalian, China), and 1µL of template DNA (100ng/µL) under the following conditions: initial denaturation of 5 min at 94°C, followed by 35 cycles (30 s at 94°C, 30 s at optimum annealing temperature, and 1 min at 72°C), as well as a final extension step at 72°C for 7 min.

Amplicons were run on 1% agarose gel to confirm the successful amplification of PCR products, and then were purified with QIAquick PCR purification kits (Qiagen) according to the manufacturer's recommendations. The purified PCR products were sequenced in both directions with the ABI 3730xl DNA Analyzer at Sangon Biotech Co. Ltd (Shanghai, China). Sequences were analyzed and edited with the Lasergene Seqman Pro 7.1.1 (DNASTAR) for consensus sequences from both forward and reverse DNA strands. All these nucleotide sequences were deposited in GenBank under the accession numbers for 167 COI (accession number: PP526771-PP526937), 109 COII (accession number: PP546395- PP546503), and 137 ITS1 (accession number: PP542683- PP542819) sequences.

## Genetic diversity and structure

The genetic diversity parameters, including the number of variable sites (S), the count of haplotypes (h), nucleotide diversity (π), haplotype diversity (Hd), the mean number of nucleotide differences (k), Tajima's D statistic, Fu and Li's D and F tests, and Fu's Fs test, were computed utilizing DnaSP version 5.10 [18]. The genetic distance matrix (FST) and the Analysis of Molecular Variance (AMOVA) were both computed using Arlequin 3.5 software [19], with variance components within and among populations estimated through 1000 random permutations to elucidate the genetic structure.

The data for the haplotype network was calculated from DnaSP6 using COI, COII, and ITS1 sequences, respectively. It was then imported and visualized by PopArt 1.7 software [20]. The network was constructed using the median joining method, and the nodes and labels of the graph were adjusted to prevent them from being obscured.

After performing a Mantel correlogram analysis and Mantel test with 1,000 permutations, using Spearman's rank correlation to examine the associations between Fst values and geographical features such as altitude, geographical distance, longitude, and latitude in *N. castanea*, as conducted via the vegan package [21] and visualized by R software.

## Phylogenetic analysis of *N. castanea* population

The COI, COII, and ITS1 sequences were initially aligned with high precision using the MAFFT7 software [22]. Subsequently, the aligned sequences were input into IQ-TREE2 [23] for the construction of a maximum likelihood (ML) phylogenetic tree with the following parameters: 1000 ultrafast bootstrap replicates were performed, the number of threads was automatically determined, and the best-fit model of sequence evolution was automatically selected based on the Bayesian Information Criterion (BIC). The resulting tree was then rendered and visualized using the iTol online tool (https://itol.embl.de/) with annotation of sites.

*Odoiporus longicollis*, along with *N. castanea*, both belong to the Curculionidae family and can be used as outgroups for comparison with *N. castanea*. The COI, COII, and ITS gene sequences of these species can be obtained from the National Center for Biotechnology Information (NCBI).

## Results

### Molecular characterization of COI, COII and ITS1

Sequences with low DNA quality were partially amplified, which was discarded without further investigation. Totally, the COI gene was successfully amplified in 170 *N. castanea* individuals, the COII gene in 109 individuals, and the ITS1 gene in 143 individuals.

For the concatenated COI and COII gene, although the result of Tajima's D was not significant, both Fu and Li's D and Fu and Li's F exhibited significant negative values (Table 1), indicating that these statistics are more sensitive to population expansion events, particularly shortly after the expansion event. Additionally, the significant negative value of Fu's Fs statistic further confirmed the hypothesis of population expansion. These results suggest that the population may have undergone complex historical events, such as a bottleneck effect following expansion or gene flow, which could have caused Tajima's D to fail to detect significant deviation, whereas the Fu and Li's statistics significantly revealed the signal of population expansion.

On the other hand, for the ITS1 marker, although the results of Fu and Li's D and Fu and Li's F were not significant, the significant negative value of Fu's Fs statistic strongly indicated a history of population expansion (Table 1). This is consistent with the significant negative value result of Tajima's D, collectively pointing to the possibility of population expansion.

### Phylogenetic relationship and haplotype network of *N. castanea* populations

**Phylogenetic relationship and haplotype network based on COI and COII sequences.** We identified a total of 50 haplotypes from concatenated COI and COII gene fragments. The

**Table 1. Number of variable sites (S), haplotypes (h), nucleotide diversity ($\pi$), haplotype diversity (Hd), the average number of nucleotide differences (k), Tajima's D, Fu and Li's D, Fu and Li's F, Fu's Fs statistic based on ITS1 and concatenated COI and COII gene fragments of *N. castanea*.**

|  | COI+COII | ITS1 |
|---|---|---|
| Sample Size | 99 | 137 |
| S | 133 | 179 |
| h | 50 | 53 |
| $\pi$ | 0.01467 | 0.00387 |
| Hd | 0.947 | 0.66 |
| k | 19.081 | 4.82 |
| Tajima's D | -0.95ns | -2.834*** |
| Fu and Li's D | -4.61915* | -7.87159 ns |
| Fu and Li's F | -3.65909* | -6.66705 ns |
| Fu's Fs statistic | -7.371*** | -34.252*** |

* *P* < 0.05

** *P* <0.01

*** *P* < 0.001

ns Not significant *P* > 0.05

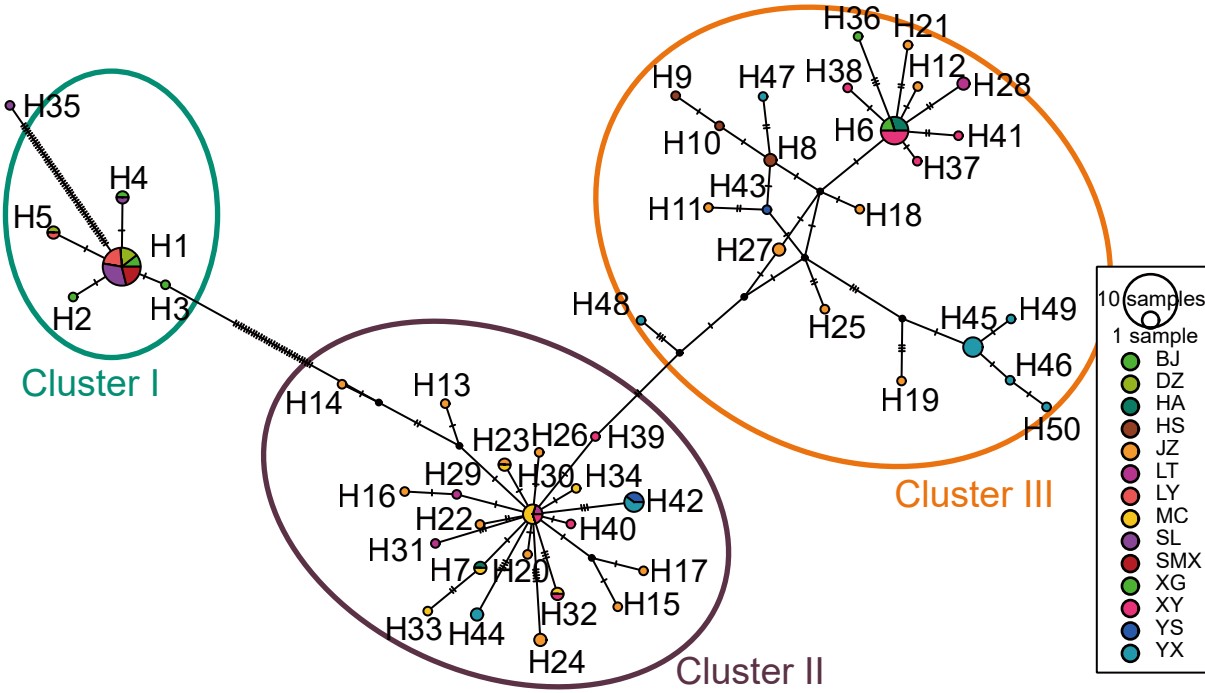

**Fig 1. Median-joining network based on the COI and COII gene haplotypes.**

generated Maximum Likelihood tree by iqtree2 was dispersed into three main distinct cluster and each clade evolved separately having a different genetic structure for each (Fig 1). The cluster I contained 6 haplotypes (H1, H2, H3, H4, H5 and H35), consistent with clade I of the phylogeny tree also clustered together (Fig 2), and they were all distributed in the western sampling points, all having relatively high elevations (Fig 3 and S1 Table). Although H45, H46, H49, and H50 belong to cluster III, they differ from other haplotypes and are found in western sampling regions with similar elevations, suggesting that altitude may influence the formation of haplotypes between populations. In addition, we found that haplotypes within Cluster II were more commonly found in the eastern sampling sites. This distribution pattern suggests that these haplotypes may be influenced by geographical distance.

**Phylogenetic relationship and haplotype network based on ITS1 sequences.**   In the ITS1 ML tree and haplotype network composed of 53 haplotypes, the distribution is relatively scattered with no distinct clustering observed (Figs 4 and 5). In the ITS1 haplotype network (Fig 4), most haplotypes are grouped into one major branch. However, there are several small branches in Fig 4 that warrant attention, such as H22 and H52 clustering into a small twig, H14, H21, H28, and H53 forming another small twig, and H2, H7, and H44 grouping together, as well as H3, H16, and H25 (Fig 5). These clusters are consistent with the results of the haplotype network (Fig 5). Compared to the networks based on other genes, the haplotype network constructed from ITS1 sequences has the smallest number of clusters. More haplotypes were exclusively associated with H24, appearing only at the JZ site of Anhui province, and not shared with other regions, suggesting its prevalence within the population, as further indicated by the COI and COII analyses in Fig 2.

## Determination of genetic diversity differentiation

Fst values for the *N. castanea* species complex reveal distinct genetic differentiation patterns among populations based on COI, COII, and ITS1 sequences. The highest Fst value for COI

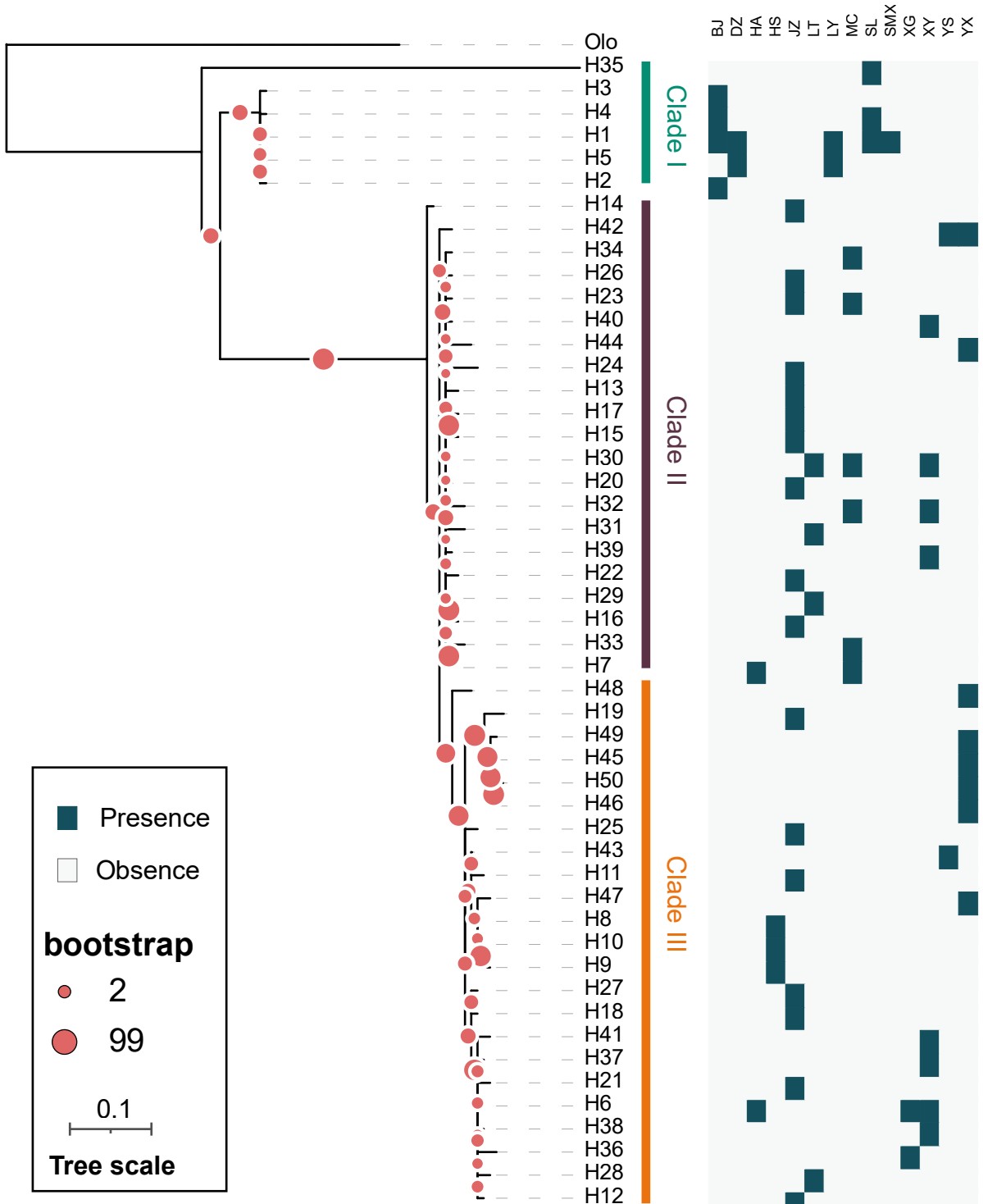

**Fig 2. A maximum likelihood phylogeny tree based on the COI and COII gene haplotypes.** *Odoiporus longicollis* was set as outgroup.

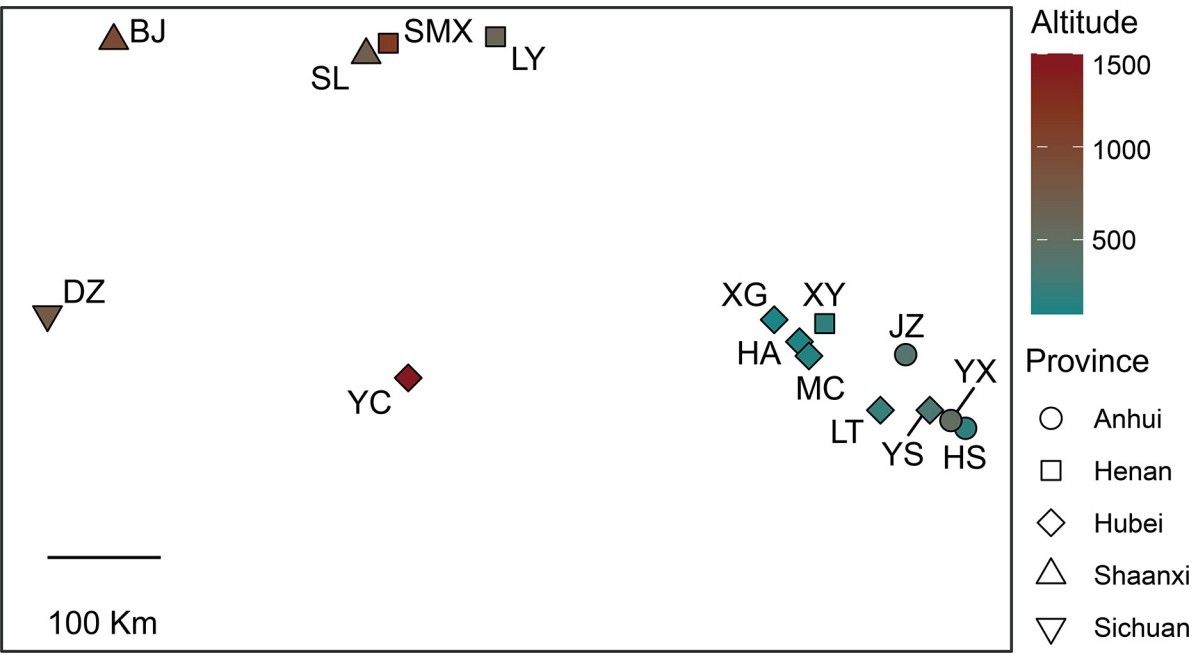

**Fig 3. Sampling locations of *N. castanea*.** The various shapes denote different provinces, where red symbols indicate high-altitude areas, and dark cyan symbols indicate low-altitude regions.

and COII sequences was observed between HS-SMX, while LY-DZ showed effectively zero differentiation (S2 Table). Similarly, the highest FST value for ITS1 sequences (0.98591) was between YC-XY, with XY-XG showing no genetic subdivision (S3 Table). The maximum Fst values were observed between sampling sites in the east and west, while the minimum Fst values were found within the eastern and western regions, suggesting that geographical isolation impacts genetic differentiation among these places.

The AMOVA analyses across the COI, COII, and ITS1 gene loci consistently revealed significant genetic differentiation among populations (Table 2). As indicated in Table 2, the genetic variation among populations for COI and COII accounted for 75.66% ($F_{ST}$ = 0.75661, $P < 0.001$) among populations, with the remaining 24.34% of variation occurring within populations, indicating substantial genetic differentiation between populations. The ITS1 locus exhibited the significant differentiation between sites (Fst = 0.09445, $P < 0.001$), with 9.45% of genetic variation among populations, and only 90.55% within populations, confirming that there were populations associated with distances among populations or different *N. castanea* regions.

## Correlation analyses of genetic differentiation and geographical data

Upon executing a Mantel correlogram analysis to examine the associations between Fst values and geographical features such as altitude, geographical distance, longitude, and latitude in *N. castanea*, the results are shown in Fig 6. Genetic differentiation, as indexed by the mtDNA (COI and COII) and ITS1 markers, significantly correlated positively with both altitude and geographical distance at lower elevations and shorter spans (Fig 6A, 6B, 6E, 6F), and interestingly, this positive correlation reversed to a significant negative correlation at moderate to high altitudes and geographical ranges for all three markers. This pattern was also observed in latitude and longitude data of mtDNA (Fig 6C and 6D), but not in ITS (Fig 6G and 6H).

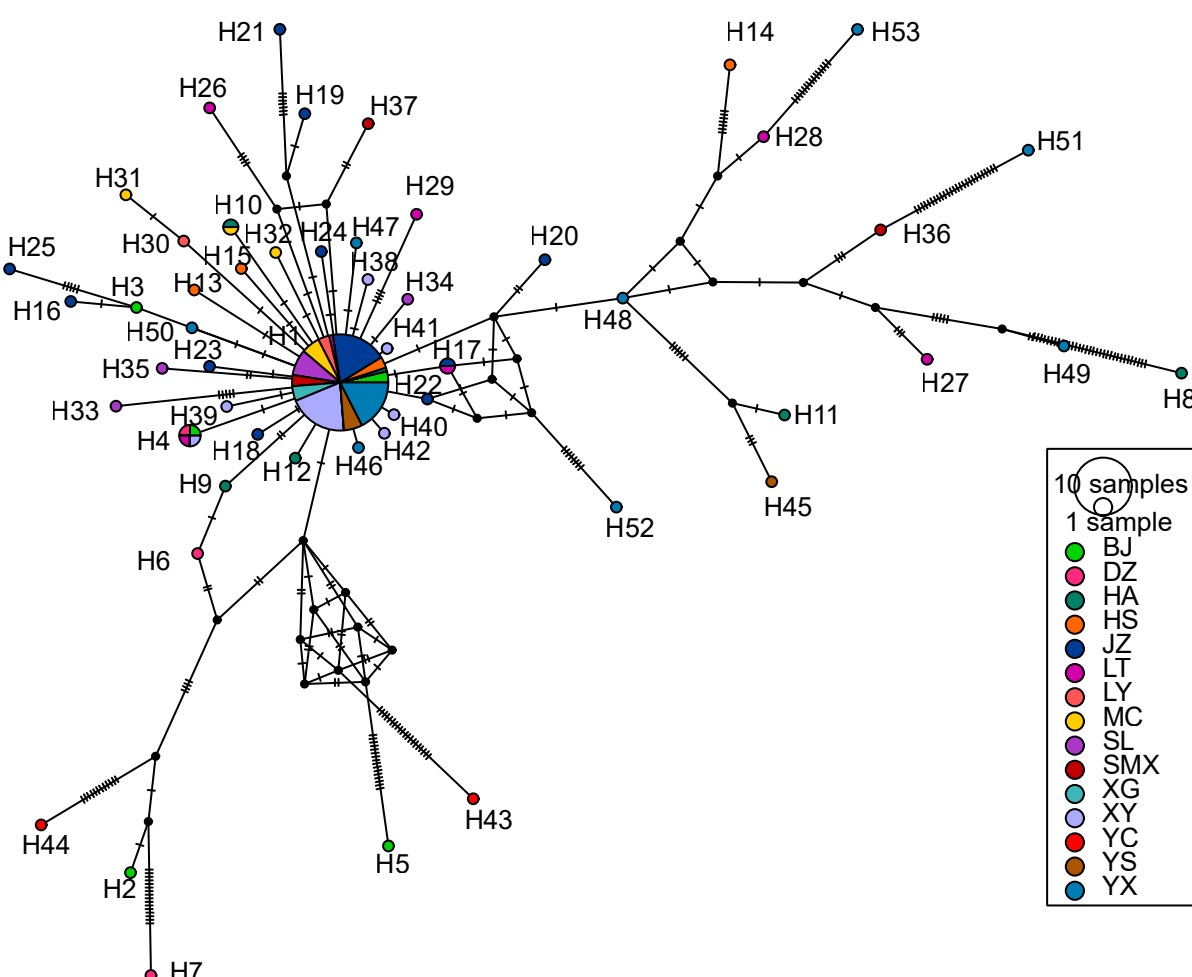

**Fig 4. Median-joining network based on the ITS1 gene haplotypes.** *Odoiporus longicollis* was set as outgroup.

## Discussion

The concurrent application of COI, COII, and ITS1 in the genetic analysis of *N. castanea* offers a holistic perspective on the species' diversity and evolutionary trajectory. Mitochondrial genes COI and COII, with their moderate evolutionary rates, serve as robust markers for deciphering phylogenetic relationships and geographic variation patterns [13, 24, 25]. In contrast, the nuclear marker ITS1, despite its high variability, likely due to a larger effective population size and varying selective pressures, provides insights into the species' evolutionary dynamics and potential selective forces [24]. Three different genetic markers (COI, COII and ITS1) both provided evidence of population expansion, although there were differences in the significance of their respective statistics. These results suggest that the studied population may have undergone an expansion event, and this event may have been accompanied by other complex historical processes, such as bottleneck effects or gene flow. Therefore, using multiple genetic markers and statistical methods can provide a more comprehensive and accurate understanding when assessing population history.

The genetic differentiation of *N. castanea* populations is influenced by their geographic location, as evidenced by the pest's presence across a broad range of latitudes and altitudes in

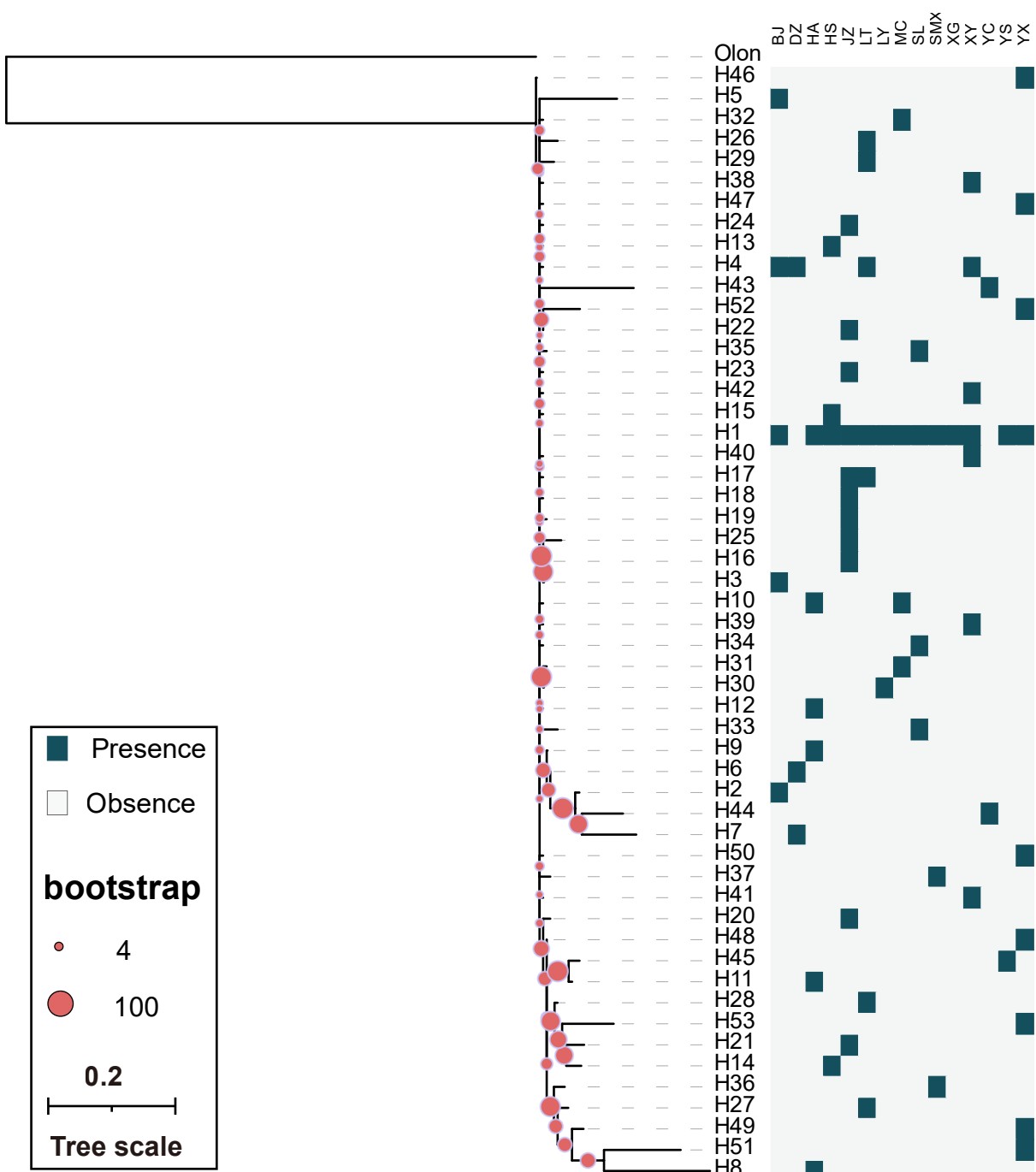

**Fig 5. A maximum likelihood phylogeny tree based on the ITS1 gene haplotypes.**

the central part of China, leading to significant genetic differences among populations, which can be attributed to the varying climatic conditions, topography, and ecological niches encountered in different regions [6, 26]. The genetic diversity within the species is essential for its adaptation to different environments and the ability to respond to environmental changes [27, 28]. A notable genetic difference was observed between *N. castanea* populations in the

**Table 2. AMOVA for genetic structure of *N. castanea* populations from China based on mtDNA and ITS fragment sequences.**

| Source of variation | Marker | Among populations | Within populations | Total |
|---|---|---|---|---|
| df | COI+COII | 13 | 85 | 98 |
| | ITS1 | 14 | 122 | 136 |
| Variance components | COI+COII | 6.48 | 2.08 | 8.56 |
| | ITS1 | 0.25 | 2.40 | 2.65 |
| Percentage variance | COI+COII | 75.66 | 24.34 | |
| | ITS1 | 9.45 | 90.55 | |
| Fixation indices (*P* value) | COI+COII | $F_{ST} = 0.76$ (*P* = 0.0000) | | |
| | ITS1 | $F_{ST} = 0.094$ (*P* = 0.0029) | | |

central and eastern regions of China in this study, which can be attributed to the distinct environmental conditions. The central region, with its mountainous terrain and diverse ecological zones [29, 30], likely provides a more complex and diverse environment for *N. castanea*, leading to increased genetic variation. In contrast, the eastern region, with its flat terrain and more homogeneous ecological conditions [29, 30], may result in a more uniform genetic composition of the species. Moreover, these genetic variations, when combined with China's geographic environment and altitude distribution, provide insights into the factors influencing population variation of *N. castanea* [6]. As we found that diversity of *N. castanea* was also affected by altitude. As it is known that the variation in altitude and climate across China creates a mosaic of ecological conditions [6, 30], which in turn may shape the genetic diversity of

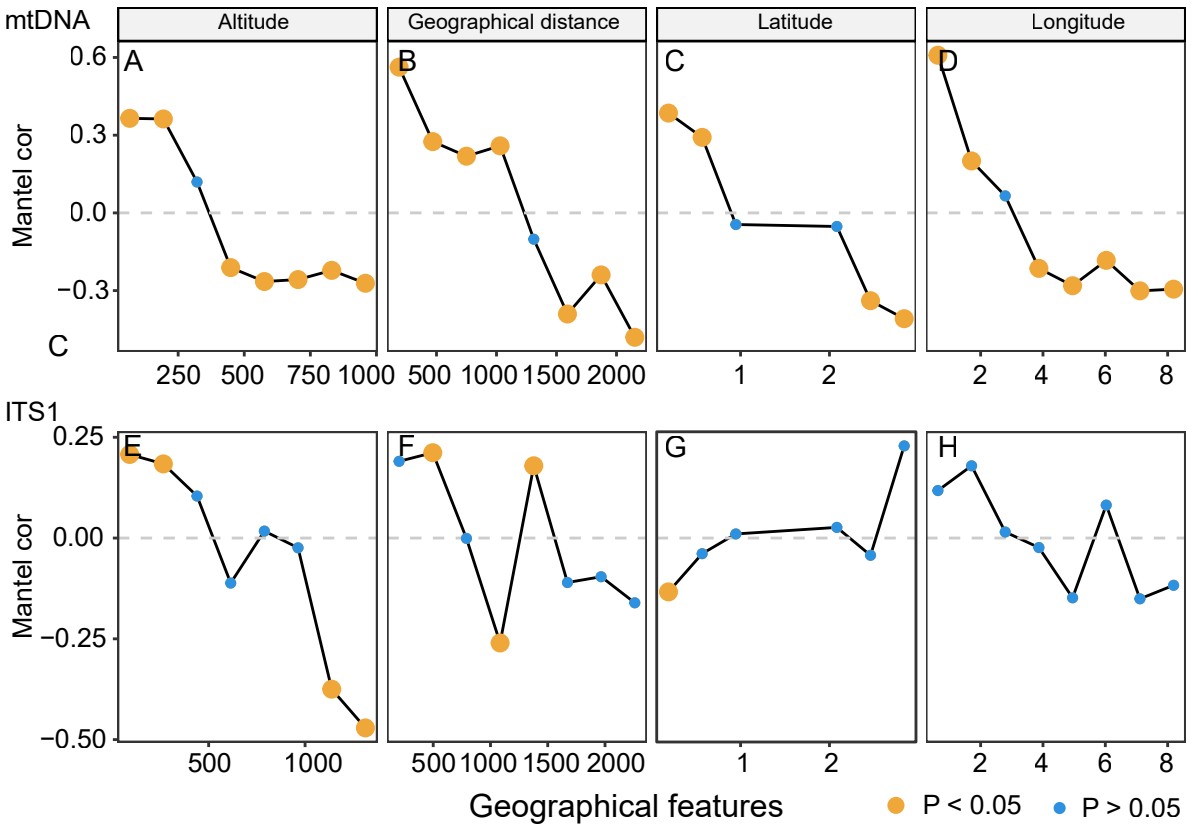

**Fig 6. Mantel Correlogram for genetic and geographic data.**

*N. castanea.* The species' ability to adapt to these diverse conditions is likely facilitated by the genetic variability present within its populations [28]. Moreover, the haplotypes of the COI and COII gene cluster more frequently in neighboring provinces, which may indicate a higher level of gene flow between these regions, supporting genetic connectivity between different areas. Some clusters are primarily located in relatively high-altitude regions, suggesting that the species in these areas may share similar ecological niches, reflecting the common influence of environmental conditions on species adaptability.

We further investigated the correlations between population genetic differentiation and geographic factors, including additional spatial variables such as altitude, longitude, and latitude. These variables exhibit complex and dynamic interactions between environmental gradients and population genetic structure. The observation of significant positive correlation between genetic differentiation and altitude in low-altitude regions may be attributed to environmental heterogeneity and the availability of diverse niches that promote genetic differentiation [26] within an ecologically suitable environment in these regions. The significant negative correlation at moderate to high altitudes for all three markers that populations separated by a certain distance were likely to be more genetically distinct [31].

This phenomenon is evident not only in relation to altitudes but also in relation to geographical distance. At lower elevations and shorter distances, geographical barriers may have constrained gene flow, resulting in a positive correlation between genetic differentiation and altitude as well as geographical distance. However, as elevation and geographical distance increase, other factors such as niche differentiation, natural selection, or historical events may come into play, leading to a negative correlation between genetic differentiation and these geographical features. This may be analogous in *Drosophila*, where genetic similarity has been detected between populations situated at a distance, likely due to similar environmental conditions; over time, the low frequency of intermixing through migration or gene flow has resulted in genetically distinct populations across their range, maintaining a high degree of structural differentiation and a semi-isolated status [31, 32]. As is widely known, Drosophila spread to other areas through the transportation of fruit. Spatial expansions can generate allele frequency gradients, promote the surfing of rare variants into newly occupied territories, induce the structuring of newly colonized areas into distinct sectors of low genetic diversity, or lead to massive introgression of local genes into the genome of an invading species [32, 33]. However, the ITS1 marker showed no significant relationship with longitude and latitude, further indicating that the genetic structure of *N. castanea* may be influenced to some extent by non-geographical factors [6].

As previously reported, human activities may have significantly influenced the distribution of *N. castanea. N. castanea* can reside within the fruits of chestnuts, and the dispersal of chestnuts to new areas as a result of human economic endeavors has augmented the distributional range of *N. castanea*, similar to fruit flies [6, 34]. However, despite the distribution of chestnuts being a significant factor in the spread of *N. castanea*, it is important to note that prior studies have underscored the lack of a direct relationship between the geographic distribution and genetic diversity of *N. castanea* [35]. The precise origin of the chestnut trees that host *N. castanea* pests remains unclear, suggesting that the spread of these populations is likely linked to chestnut transportation [16, 36], and the difficulty in detecting the pest within the chestnut fruit during harvest [16] further increases the probability of its transfer. Human planting activities may have provided new habitats and food resources for the pest, thereby promoting its dispersal and establishment of new populations. This could have, in part, affected the genetic diversity and gene flow within local populations of *N. castanea*. Moreover, the extensive distribution of chestnut plants, coupled with the use of insecticides, has substantially influenced the distribution of *N. castanea* [16]. The effects of habitat fragmentation, pesticide use, and

agricultural practices have significantly influenced the distribution and abundance of *N. castanea* populations [37]. On one hand, the variation observed reflects the species' adaptive and evolutionary responses to a range of geographical features. On the other hand, the occurrence of parasitic infestations by pests within chestnuts, which aids in their dispersal to new regions via chestnut transportation, demands the enhancement of monitoring and control measures during transit to prevent the geographical spread of these pest populations. Comprehending these impacts is essential for devising effective management strategies that balance pest control requirements with the preservation of biodiversity.

## Conclusion

In summary, the distribution of *N. castanea*, is intricately linked to ecological and geographical features. The genetic variation of *N. castanea* is influenced by geographic location, with significant differences observed between populations in central and eastern China. Our study reveals a complex relationship between genetic variation and spatial parameters, with non-linear patterns observed at intermediate distances and altitudes. The use of COI and COII genes as genetic markers has proven effective in inferring genetic diversity and evolutionary relationships, while the ITS1 region's high variability can provide additional insights into the evolutionary dynamics and potential selective pressures acting on the species. These findings contribute to our understanding of the factors shaping the genetic diversity of *N. castanea* and provide insights for understanding of their genetic diversity and distribution.

## Supporting information

**S1 Table. Information on sample locations of chestnut pest *Niphades castanea*.**
(DOCX)

**S2 Table. Fixation index (Fst) values among populations based on concatenated COI and COII gene sequences of *N. castanea*.**
(DOCX)

**S3 Table. Fixation index (Fst) values among populations based on ITS1 sequences of *N. castanea*.**
(DOCX)

**S4 Table. Nucleotide sequence deposits in GenBank with accession numbers.**
(XLSX)

## Author Contributions

**Conceptualization:** YunLi Xiao.

**Data curation:** Yue Fu, JiaXin Wang, Peng Yu.

**Formal analysis:** Yue Fu, JiaXin Wang, Peng Yu.

**Funding acquisition:** YunLi Xiao.

**Investigation:** Yue Fu, JiaXin Wang, Peng Yu.

**Methodology:** Bin Mao, Mi Shen.

**Project administration:** YunLi Xiao.

**Resources:** YunLi Xiao.

**Software:** Bin Mao, Mi Shen.

**Validation:** Yue Fu, JiaXin Wang, Peng Yu.

**Visualization:** Bin Mao, Mi Shen.

**Writing – original draft:** Bin Mao, Mi Shen, YunLi Xiao.

**Writing – review & editing:** YunLi Xiao.

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
