## [Decision Letter · Decision Letter 0]

16 Jun 2024

PONE-D-24-18040Phylogeographic pattern of chestnut pest Niphades castanea Chao (Coleoptera: Curculionidae) in China based on mtDNA and ITS markersPLOS ONE

Dear Dr. Mao,

Thank you for submitting your manuscript to PLOS ONE. After careful consideration, we feel that it has merit but does not fully meet PLOS ONE’s publication criteria as it currently stands. Therefore, we invite you to submit a revised version of the manuscript that addresses the points raised during the review process.

We look forward to receiving your revised manuscript.

Kind regards,

Murtada D. Naser

Academic Editor

PLOS ONE

“This study was supported by the National Science and Technology Fundamental Resources Investigation Program of China (Grant No. 2019FY101800)”

3. We note that Figure 2 in your submission contain [map/satellite] images which may be copyrighted. All PLOS content is published under the Creative Commons Attribution License (CC BY 4.0), which means that the manuscript, images, and Supporting Information files will be freely available online, and any third party is permitted to access, download, copy, distribute, and use these materials in any way, even commercially, with proper attribution. For these reasons, we cannot publish previously copyrighted maps or satellite images created using proprietary data, such as Google software (Google Maps, Street View, and Earth). For more information, see our copyright guidelines: http://journals.plos.org/plosone/s/licenses-and-copyright.

Reviewers' comments:

Reviewer's Responses to Questions

**Comments to the Author**

1. Is the manuscript technically sound, and do the data support the conclusions?

Reviewer #1: No

Reviewer #2: Partly

Reviewer #3: Yes

2. Has the statistical analysis been performed appropriately and rigorously? 

Reviewer #1: No

Reviewer #2: Yes

Reviewer #3: N/A

3. Have the authors made all data underlying the findings in their manuscript fully available?

Reviewer #1: Yes

Reviewer #2: Yes

Reviewer #3: Yes

4. Is the manuscript presented in an intelligible fashion and written in standard English?

Reviewer #1: No

Reviewer #2: No

Reviewer #3: Yes

5. Review Comments to the Author

Reviewer #1: The paper studied genetic variance of the chestnut pest Niphades castanea in China. The authors genotyped 170 bettles from several locations.

The paper suffers from lack of scope. The aim formulated by authors was to analyse genetic diversity of the N. castanea " hoping to supply evidence for developing scientific strategies for prevention and control", yet neither results nor discussion addresses this aim.

In my opinion the paper should be considerably shortened. There are many paragraphs that add nothing new to the paper. For instance, second paragraph of the Introduction can be removed, as it states generally known facts about the genetic variance in general. Similarly, the first paragraph of the Discussion is very general and does not add any specific information relevant to the scope of the study.

I am also concerned about the finding of the neutrality tests. In particular, the Tajima's D is sensitive to demographic processes and with any "external" information about processes in the population it is not possible to state whether negative D resulted from purifying selection or population expansion. Any interpretation of the results should be treated with caution. The authors may refer to Pentinsaari et al. 2016 DOI: 10.1038/srep35275

Writing needs some improvement. E.g. in the first paragraph of the Introduction, words as "important" or "particular" are unnecessarily repeated.

Most of the results (e.g. section 3.1, 3.2 and most of the section 3.3) could be summarized in one or two sentences. Authors unnecessary repeat information from tables, which btw should be placed as Supplementary. They also provide many unimportant details such as AC/GC ratio. The only part of the results that really provides information relevant to the aim of the study are haplotype networks and section 3.5. Thus, the results should be revised and considerably shortened.

On the other hand, the core results are not adequately presented. I could not find AMOVA table nor the description. The Fig 6 is hard to read, I suggest providing a classic Fst matrix instead.

I'm not convinced that the selection of the markers was right for the study. In line the authors wrote "ITS1 is a very conserved

gene, which was not suitable to distinguish N. castanea populations at the species levels in this study" which questions its use in the paper.

An important question which is not covered in the discussion is the human impact on the population structure studied in the paper. Neither Introduction nor the Discussion mention that. Yet, I imagine that in the case of a cultivated species, human activities may be a main driver of the population structure. How likely it is that the pests were also transferred between locations by humans e.g. with a harvest? Is the origin of the host trees known? I would expect that at least some of the populations were planted by humans.

Section 2.1. A map of the locations would be helpful. The table itself can go to the Supplementary materials.

Section 2.4. Were sequences from N. castanea from other locations (outside China) included? Why was Dorysthenes paradoxu selected as an outgroup?

Section 3.1 The first paragraph of the Results is in fact description of the study site and should be moved to the Methods.

Section 3.2 is obsolete. Is does not provide any useful information given the goal of the paper.

Table 3. Why were selection tests calculated?

Table 4-6. Should be moved to the Supplementary materials, as the result is better pictured in haplotype networks

Figure 4B and 5B. Why there are some haplotypes not connected to the network?

Figure 6 is unnecessarily complicated. Instead, I suggest presenting classic Fst matrix between studied sites.

66 I can't see what makes N. castanea an "excellent model" for evolutionary studies. There is no arguments supporting this statement.

96 It is not quality of a genome, as the authors didn't study genomes. A gel can visualise a quality of the DNA isolates.

132 What were the selection tests calculated? All markers (ITS and mtDNA) are expected to be selectively neutral

180 This sentence should be rephrased: "amplification products of ITS1 gene WERE FOUND in 143 N. castanea individuals.", not "existed"

181 Relatively low variability of COI (mtDNA) is typical for this gene and mitochondrial markers in general, did the authors expected otherwise?

190 There is no sense in comparing ITS and COI/COII, as one is a nuclear gene, and the others are mtDNA. Simply because of that they are expected to differ in the levels of polymorphisms.

194 I don't think describing AC to GC ratio provides any useful information.

216 Please, don't use "pops" instead of "populations".

348-353 Markers used in the study have well described properties. I see no sense in reporting this as something "find" in the paper. This is was should be known and expected from the very beginning.

354 There is no data supporting this claim. The paper did not provide any data to test whether the changes were "historical" (what I understand as "happened during human presence in the area") or happened before.

Reviewer #2: Mao et al. present an analysis of population genetic structure for Niphades castanea a pest of chestnut. They use mitochondrial COI and COII and nuclear ITS1 region to estimate the relationships among 15 populations. They found population genetic structure based on the mitochondrial genes but little for the ITS1 locus. These are typical patterns found with other beetle population genetic studies. The methods are appropriate and the results are explained in detail. However, the discussion is shallow.

A major issue with the analyses is that the author examined the mitochondrial genes separately. Why? The genes should only differ in the rates of nucleotide substitution but not overall evolution- these genes are inherited as a single unit in the mitochondrial genome. Describing the differences in rates of nucleotide substitution is OK but haplotypes, geneflow, and phylogeny should have been analyzed as one unit including the COI and COII genes. This manuscript would be greatly improved by including analyses that combine these two genes.

Another major issue is the discussion which basically repeats the results and gives some reference to the neutrality of genetic loci. However, the authors do not discuss these results in comparison to other beetle phylogeographic studies in or out of China. This type of discussion would help readers understand if the patterns observed for Niphades castanea are typical or deviant. Adding this type of discussion would greatly improve the manuscript.

Specific issues:

1. There are typographical errors throughout the manuscript. Please review again. Be sure all scientific names are italic. Also, italics are used throughout the manuscript for non-Latin words which I assume is for emphasis. I would advise not to do this but it is a decision for the editor.

2. Maximum parsimony and maximum likelihood are confusedly used in some instances:

Line 146: “…performed till no shorter equally parsimonious trees were obtained.” Like the authors state in the above sentences, IQ-Tree produces maximum likelihood trees thus the tree search would end with a maximum likelihood tree (or set of trees), not most parsimonious trees.

Lines 375-376: “The analysis of COI and COII sequences revealed consistent results between

Maximum Parsimony (MP) and Neighbor-Joining (NJ) analyses.” The use of maximum parsimony was not detailed in the Materials and Methods.

3. Lines 349 -352: “Nuclear marker ITS1 displayed more variability, diversity, GC content, average evolutionary divergence over sequence pairs, and lower indel variability than plastid markers COI and COII. However, ITS1 is a very conserved gene…”. If ITS1 is more variable then how is it more conserved?

4. Table 4- “Information on the COI haplotype of COII shared within different pops” This title does not make sense. Perhaps delete table with given the combine analysis of COI and COII will make this table unnecessary?

Reviewer #3: My major concern lies in the phylogenetic analyses in this study.

The focal tree should rely on the model-based ML analyses, but the author seemed to stick to the out-of-date analyses based on parsimony and NJ tree.

Line 15-16， fruit insect pest of chestnuts (Castanea spp.), Needing rewording.

Lines 36, 379: Castanea and N. castanea in italics

Line 81, ‘N.’ should in full generic name here.

Iine 143, by IQ-TREE (Yu et al. 2020) with maximum likelihood (ML) analysis. Please add details about the parameters, model selection, and model used in the phylogenetic analyses. You cannot only say ‘evaluated by IQ-TREE’; the readers need to know the details.

Line 375, Maximum Parsimony (MP) and Neighbor-Joining (NJ) analyses were compared, but how about the ML method using IQTREE for these gene markers? You should also discuss the differences and why.

6. PLOS authors have the option to publish the peer review history of their article (what does this mean?). If published, this will include your full peer review and any attached files.

Reviewer #1: **Yes: **Agnieszka Kloch

Reviewer #2: No

Reviewer #3: No

---

## [Author Response · Author response to Decision Letter 0]

2 Sep 2024

Review Comments to the Author

Reviewer #1: The paper studied genetic variance of the chestnut pest Niphades castanea in China. The authors genotyped 170 bettles from several locations. The paper suffers from lack of scope. The aim formulated by authors was to analyse genetic diversity of the N. castanea " hoping to supply evidence for developing scientific strategies for prevention and control", yet neither results nor discussion addresses this aim.

Re: We conducted sampling at sites where the pest has been reported throughout China, and in some regions, the presence of the pest was not detected in the Materials sampling section (line 68-79). For pest prevention and control, we have reanalyzed the scope as outlined in both the Introduction section (lines 43-47) and the Discussion section (lines 344-370) to enhance clarity. The revisions have been highlighted in red.

In my opinion the paper should be considerably shortened. There are many paragraphs that add nothing new to the paper. For instance, second paragraph of the Introduction can be removed, as it states generally known facts about the genetic variance in general. Similarly, the first paragraph of the Discussion is very general and does not add any specific information relevant to the scope of the study.

Re: The second paragraph of the Introduction has been streamlined to include only content relevant to the topic, with the generally known facts about genetic variance in general having been removed. Additionally, the Discussion section has been rewritten.

I am also concerned about the finding of the neutrality tests. In particular, the Tajima's D is sensitive to demographic processes and with any "external" information about processes in the population it is not possible to state whether negative D resulted from purifying selection or population expansion. Any interpretation of the results should be treated with caution. The authors may refer to Pentinsaari et al. 2016 DOI: 10.1038/srep35275

Re: We integrated the results from Tajima’s D statistic, Fu and Li’s D and F tests, and Fu’s Fs test, and considering the species' frequent translocation in economic trade, we interpret these findings as indicative of population expansion. The results in this section have been reorganized and are now presented on lines 151-164.

Writing needs some improvement. E.g. in the first paragraph of the Introduction, words as "important" or "particular" are unnecessarily repeated. Most of the results (e.g. section 3.1, 3.2 and most of the section 3.3) could be summarized in one or two sentences. Authors unnecessary repeat information from tables, which btw should be placed as Supplementary. They also provide many unimportant details such as AC/GC ratio. The only part of the results that really provides information relevant to the aim of the study are haplotype networks and section 3.5. Thus, the results should be revised and considerably shortened. On the other hand, the core results are not adequately presented. I could not find AMOVA table nor the description. The Fig 6 is hard to read, I suggest providing a classic Fst matrix instead. 

Re: We have rewritten the first paragraph of the Introduction and reanalyzed the Results section, with both sections now highlighted in red. To sharpen the focus of the text, extraneous information has been eliminated. Furthermore, the results from the AMOVA analysis have been reintegrated into the Results section, appearing at lines 224-232. Additionally, the haplotype networks have been reanalyzed and are presented at lines 180-191 and 198-209. The original Figure 6 has already been replaced with the Mantel test, and the Fst matrix can be found in S2 to S4 Tables.

I'm not convinced that the selection of the markers was right for the study. In line the authors wrote "ITS1 is a very conserved gene, which was not suitable to distinguish N. castanea populations at the species levels in this study" which questions its use in the paper.

Re: We deeply apologize for the imprecision in our previous statements. The conservation of ITS sequences is mainly observed among populations, with considerable variation observed within populations. However, in this study, 9.45% of the genetic variation is still significant (P < 0.05) at line 224-232. We have reorganized the data and reanalysis of the ITS1 sequences. Appropriate revisions have been made in the Results and Discussion section, and these changes are all highlighted in red.

An important question which is not covered in the discussion is the human impact on the population structure studied in the paper. Neither Introduction nor the Discussion mention that. Yet, I imagine that in the case of a cultivated species, human activities may be a main driver of the population structure. How likely it is that the pests were also transferred between locations by humans e.g. with a harvest? Is the origin of the host trees known? I would expect that at least some of the populations were planted by humans.

Re: Thank you for your valuable suggestions. We have addressed the impact of human activities in the Introduction and Discussion section, with the analysis presented at lines 43-47 and line 350-324. The pest N. castanea is known to inhabit the chestnut fruit, making it difficult to detect during harvest, which consequently leads to a high probability of its transfer to new locations. This transfer is likely facilitated by human activities, as the precise origin of the chestnut trees that host these pests remains unclear. Human planting activities may have provided new habitats and food resources for the pest, thereby promoting its dispersal and establishment of new populations. Considering this, we will revise the discussion sections of our manuscript. 

Section 2.1. A map of the locations would be helpful. The table itself can go to the Supplementary materials.

Re: Considering the time-consuming process of the map application and approval process, we have developed Figure 1 using a method that relies on the latitude and longitude of the sampling points. The original Table 1 has been included as an attachment S1 Table.

Section 2.4. Were sequences from N. castanea from other locations (outside China) included? Why was Dorysthenes paradoxu selected as an outgroup?

Re: Regarding the choice of outgroup, we used the blastn method to compare the COI, COII, and ITS1 sequences. From the comparison results, we selected the corresponding sequences of the species with the highest similarity, whose the COI, COII, and ITS gene sequences are known, ensuring that these sequences have a close phylogenetic relationship with the target sequences, to be used as the outgroup for constructing the phylogenetic tree. Based on this criterion, Odoiporus longicollis was chosen as the outgroup for reconstructing the evolutionary trees for the three genes (COI, COII, and ITS1). This content has been added to the Methods section, lines 140-143.

Section 3.1 The first paragraph of the Results is in fact description of the study site and should be moved to the Methods.

Re: The first paragraph of the Results has been moved to the Methods section.

Section 3.2 is obsolete. Is does not provide any useful information given the goal of the paper.

Re: This paragraph has been deleted.

Table 3. Why were selection tests calculated?

Re: The original Table 3 has been removed.

Table 4-6. Should be moved to the Supplementary materials, as the result is better pictured in haplotype networks

Re: Since the haplotype networks have fully encompassed the content of Tables 4-6, we have deleted Tables 4-6.

Figure 4B and 5B. Why there are some haplotypes not connected to the network?

Re: This issue may be due to an error in parameter settings. We have redrawn the haplotype networks in Figures 2, and 4 using the PopArt software.

Figure 6 is unnecessarily complicated. Instead, I suggest presenting classic Fst matrix between studied sites.

Re: The Fst matrix between the studied sites is presented in Table S2 and S3.

66 I can't see what makes N. castanea an "excellent model" for evolutionary studies. There is no arguments supporting this statement.

Re: The sentence has been revised at line 55-56.

96 It is not quality of a genome, as the authors didn't study genomes. A gel can visualise a quality of the DNA isolates.

Re: The sentence has been revised at line 84.

132 What were the selection tests calculated? All markers (ITS and mtDNA) are expected to be selectively neutral

Re: We acknowledge that both ITS and mtDNA are generally considered to be selectively neutral markers. However, the purpose of applying Tajima’s D and Fu’s FS tests was not to directly test for selection, but rather to confirm the assumption of neutrality for these markers in our dataset, ensuring that any patterns observed are not due to non-neutral processes.

180 This sentence should be rephrased: "amplification products of ITS1 gene WERE FOUND in 143 N. castanea individuals.", not "existed"

Re: The paragraph has been revised at 148-150.

181 Relatively low variability of COI (mtDNA) is typical for this gene and mitochondrial markers in general, did the authors expected otherwise?

Re: The sentence has been removed, and the paragraph has been rewritten on lines 151-164.

190 There is no sense in comparing ITS and COI/COII, as one is a nuclear gene, and the others are mtDNA. Simply because of that they are expected to differ in the levels of polymorphisms.

Re: The paragraph has been revised, and the descriptive parts have been corrected in lines 151-164. Our intention in including both nuclear and mitochondrial markers in our study was to provide a more comprehensive understanding of the genetic diversity and evolutionary relationships within the species under investigation. While it is true that nuclear and mitochondrial genes can exhibit different levels of polymorphism, each offers unique insights. 

194 I don't think describing AC to GC ratio provides any useful information.

Re: The corresponding section has been deleted.

216 Please, don't use "pops" instead of "populations".

Re: We have made corrections to the entire manuscript.

348-353 Markers used in the study have well described properties. I see no sense in reporting this as something "find" in the paper. This is was should be known and expected from the very beginning.

Re: The discussion section has been rewritten.

354 There is no data supporting this claim. The paper did not provide any data to test whether the changes were "historical" (what I understand as "happened during human presence in the area") or happened before.

Re: We have reanalyzed the data and rewritten the discussion section accordingly.

Reviewer #2: Mao et al. present an analysis of population genetic structure for Niphades castanea a pest of chestnut. They use mitochondrial COI and COII and nuclear ITS1 region to estimate the relationships among 15 populations. They found population genetic structure based on the mitochondrial genes but little for the ITS1 locus. These are typical patterns found with other beetle population genetic studies. The methods are appropriate and the results are explained in detail. However, the discussion is shallow.

Re: The discussion section has been rewritten.

A major issue with the analyses is that the author examined the mitochondrial genes separately. Why? The genes should only differ in the rates of nucleotide substitution but not overall evolution- these genes are inherited as a single unit in the mitochondrial genome. Describing the differences in rates of nucleotide substitution is OK but haplotypes, geneflow, and phylogeny should have been analyzed as one unit including the COI and COII genes. This manuscript would be greatly improved by including analyses that combine these two genes.

Re: We have reanalyzed the data using concatenated COI and COII gene fragments along with ITS sequences.

Another major issue is the discussion which basically repeats the results and gives some reference to the neutrality of genetic loci. However, the authors do not discuss these results in comparison to other beetle phylogeographic studies in or out of China. This type of discussion would help readers understand if the patterns observed for Niphades castanea are typical or deviant. Adding this type of discussion would greatly improve the manuscript.

Re: Limited data is available for comparison with other chestnut pests. Nevertheless, this pattern is highly reminiscent of that observed in fruit flies, which are also fruit pests. Consequently, we have undertaken a comparative analysis between the two, as detailed at 304-320.

Specific issues:

1. There are typographical errors throughout the manuscript. Please review again. Be sure all scientific names are italic. Also, italics are used throughout the manuscript for non-Latin words which I assume is for emphasis. I would advise not to do this but it is a decision for the editor.

Re: The entire manuscript has been revised.

2. Maximum parsimony and maximum likelihood are confusedly used in some instances:

Re: The corresponding part of the Methods section has been rewritten.

Line 146: “…performed till no shorter equally parsimonious trees were obtained.” Like the authors state in the above sentences, IQ-Tree produces maximum likelihood trees thus the tree search would end with a maximum likelihood tree (or set of trees), not most parsimonious trees.

Re: The corresponding part of the Methods and Results section has been rewritten.

Lines 375-376: “The analysis of COI and COII sequences revealed consistent results between Maximum Parsimony (MP) and Neighbor-Joining (NJ) analyses.” The use of maximum parsimony was not detailed in the Materials and Methods.

Re: In consideration of the comments from the reviewers, we have maintained the Maximum Likelihood (ML) phylogenetic tree generated by IQ-TREE2 for our analyses.

3. Lines 349 -352: “Nuclear marker ITS1 displayed more variability, diversity, GC content, average evolutionary divergence over sequence pairs, and lower indel variability than plastid markers COI and COII. However, ITS1 is a very conserved gene…”. If ITS1 is more variable then how is it more conserved?

Re: We have recognized the imprecision in our expression and have reorganized the data. Consequently, we have rewritten the Results and Discussion sections.

4. Table 4- “Information on the COI haplotype of COII shared within different pops” This title does not make sense. Perhaps delete table with given the combine analysis of COI and COII will make this table unnecessary?

Re: We have deleted the corresponding table.

Reviewer #3: My major concern lies in the phylogenetic analyses in this study.

The focal tree should rely on the model-based ML analyses, but the author seemed to stick to the out-of-date analyses based on parsimony and NJ tree.

Re: The Maximum Likelihood-based phylogenetic tree, as shown in Figures 3 and 5, has been constructed, replacing the previous evolutionary tree.

Line 15-16，fruit insect pest of chestnuts (Castanea spp.), Needing rewording.

Re: The sentence has been revised.

Lines 36, 379: Castanea and N. castanea in italics

Re: The sentence has been revised.

Line 81, ‘N.’ should in full generic name here.

Re: The sentence has been revised.

Iine 143, by IQ-TREE (Yu et al. 2020) with maximum likelihood (ML) analysis. Please add details about the parameters, model selection, and model used in the phylogenetic analyses. You cannot only say ‘evaluated by IQ-TREE’; the readers need to know the details.

Re: We have already included the tree construction parameters in the ' Phylogenetic analysis of N. castanea population' section of the methods at 131-143.

Line 375, Maximum Parsimony (MP) and Neighbor-Joining (NJ) analyses were compared, but how about the ML method using IQTREE for these gene markers? You should also discuss the differences and why.

Re: In consideration of the comments from the reviewers, we have maintained the Maximum Likelihood (ML) phylogenetic tree generated by IQ-TREE2 for our analyses.

---

## [Editor Report · Decision Letter 1]

4 Sep 2024

Unexpectedly complex distribution pattern of chestnutpest Niphades castanea Chao (Coleoptera: Curculionidae) based on mtDNA and ITS markers

PONE-D-24-18040R1

Dear Dr. YunLi Xiao,

We’re pleased to inform you that your manuscript has been judged scientifically suitable for publication and will be formally accepted for publication once it meets all outstanding technical requirements.

Kind regards,

Murtada D. Naser

Academic Editor

PLOS ONE
---

## [Editor Report · Acceptance letter]

16 Sep 2024

PONE-D-24-18040R1 

PLOS ONE

Dear Dr. Xiao, 

I'm pleased to inform you that your manuscript has been deemed suitable for publication in PLOS ONE. Congratulations! Your manuscript is now being handed over to our production team.

Kind regards, 

on behalf of

Dr. Murtada D. Naser 

Academic Editor

PLOS ONE